# Isotropy-Enhanced Conditional Masked Language Models

**Pei Guo,**[*] **Yisheng Xiao,**[*] **Juntao Li,**[†] **Yixin Ji, Min Zhang**
Institute of Computer Science and Technology, Soochow University, China
{pguolst,ysxiaoo}@stu.suda.edu.cn;
{ljt, minzhang}@suda.edu.cn;
jiyixin169@gmail.com

## Abstract

Non-autoregressive models have been widely used for various text generation tasks to accelerate the inference process but at the cost of generation quality to some extent. To achieve a good balance between inference speedup and generation quality, iterative NAR models like CMLM and Disco are proposed. Researchers have made much follow-up progress based on them, and some recent iterative models can achieve very promising performance while maintaining significant speedup. In this paper, we give more insights into iterative NAR models by exploring the anisotropic problem, i.e., the representations of distinct predicted target tokens are similar and indiscriminative. Upon the confirmation of the anisotropic problem in iterative NAR models, we first analyze the effectiveness of the contrastive learning method and further propose the Look Neighbors strategy to enhance the learning of token representations during training. Experiments on 4 WMT datasets show that our methods consistently improve the performance as well as alleviate the anisotropic problem of the conditional masked language model, even outperforming the current SoTA result on WMT14 EN $\rightarrow$ DE[1].

## 1 Introduction

Transformer-based models (Vaswani et al., 2017) with auto-regressive (AR) decoding paradigm have successfully applied to various text generation tasks, such as machine translation (Wu et al., 2019; Liang et al., 2021), text summarization (Savelieva et al., 2020; Elsaid et al., 2022), dialogue systems (Zhang et al., 2020; Ma et al., 2020), with exciting performance being achieved. However, the low inference efficiency prohibits their usage (especially with current super-large transformer models as backbones) since the AR paradigm predicts the

tokens one by one in left-to-right order. To speed up model inference, many non-autoregressive methods (Gu et al., 2018; Ghazvininejad et al., 2019; Qian et al., 2021; Huang et al., 2022b; Guo et al., 2023) explore predicting target tokens in parallel in recent years. As a result of parallel decoding, they inevitably neglect the internal dependency of the target tokens, causing the generation quality falls behind their AR counterparts to some extent.

To achieve a better trade-off between inference speedup and generation quality, many iterative NAR models (Ghazvininejad et al., 2019; Kasai et al., 2020; Chan et al., 2020; Huang et al., 2022c) have been proposed recently. They adopt multiple decoding iterations to generate final results, where the intermediate result generated from the last iteration can provide useful target information and then be fed into the model for refinements in the next iteration. Excitingly, iterative decoding brings significant performance improvements for NAR models and even outperforms the vanilla autoregressive Transformer. Among them, one of the most competitive and widely-used models is CMLM (Ghazvininejad et al., 2019), which adopts uniform masking during training and a mask-predict strategy for decoding. Based on CMLM, many advanced methods, such as training strategy (Kasai et al., 2020), criterion (Marjan et al., 2020; Du et al., 2021), and inference strategy (Kreutzer et al., 2020; Geng et al., 2021), have been explored and achieved better performance. Despite significant progress, some critical problems of NAR methods still remain challenging, e.g., repetition and reliance on distillation.

To disclose more insights into the enhancement of the iterative NAR models, this paper explores from a different perspective, i.e., the anisotropic problem, which has been revealed recently in AR models (Su et al., 2022). This can cause the degeneration of AR neural language models with the anisotropic distribution of token representations

---

[*]Equal Contribution

[†] Juntao Li is the Corresponding Author.

[1]Our code implementation and data are available at https://github.com/AllForward/Isotropy-Enhanced-CMLM

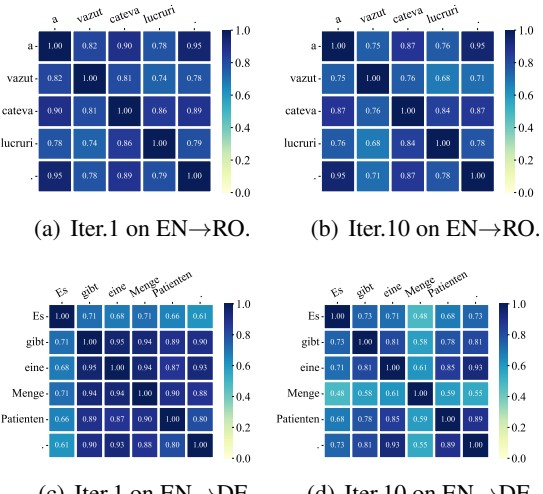

(a) Iter.1 on EN→RO.    (b) Iter.10 on EN→RO.

(c) Iter.1 on EN→DE.    (d) Iter.10 on EN→DE.

Figure 1: Token cosine similarity matrix of CMLM with in iteration 1/10 on WMT16 EN→RO and WMT14 EN→DE raw data.

(taken from the output layer of the model), in which the high similarity of different tokens can mislead the model to generate repetitive tokens at different steps. Naturally, we wonder whether the repetition problem of NAR models is also closely associated with the anisotropic representations. We take the typical iterative NAR model CMLM (Ghazvinine-jad et al., 2019), and present the results in Figure 1, the same phenomenon in AR model (Su et al., 2022) also holds for iterative NAR models, i.e., the token similarity is close to each other (Figure 1(a) and (c)). Besides, decoding with multiple iterations (Figure 1(b) and (d)) do not alleviate the dense representation of token similarity for a large margin. Therefore, the anisotropic problem of the iterative NAR models is worth more exploration.

To alleviate the anisotropic problem of CMLM, we first test the effectiveness of the existing method based on contrastive learning (Su et al., 2022). However, we found that directly adopting the existing method does not obtain consistent improvements on various datasets. Then, we analyze the potential reason and attribute it to the failure to learn the token representation well during training, which is a widely recognized problem for all NAT models. The reason why "failure to learn representations" is not conducive to improving performance will be discussed in Section 2.3. Motivated by the particularly dense representation of adjacent tokens, we propose a simple yet effective strategy named Look Neighbors to enhance the dependency of adjacent tokens and auxiliarily learn better repre-

sentations. Due to learning better representations, Look Neighbors can supplement contrastive learning well during the training of CMLM.

We conduct comprehensive experiments to evaluate our methods. Results on WMT14 EN↔DE and WMT16 EN↔RO datasets all demonstrate that our methods outperform the vanilla CMLM model, achieve comparable performance compared with strong iterative NAR baselines. Besides, the explorations on the token representation of CMLM also give a new insight into the iterative NAR models, indicating the anisotropic problem, which has been well learned in AR models recently, also affects the performance and further developments for NAR models.

## 2    Preliminaries and Trials

### 2.1    Non-autoregressive Language Model

Most of the text generation models adopt the autoregressive decoding format to generate the target sentences and use the autoregressive factorization to maximize the following likelihood:

$$\mathcal{L}_{\text{AR}} = \sum_{t}^{T} \log P(y_t | y_{<t}, X; \theta), \quad (1)$$

where $y_{<t}$ denotes the previous generated target tokens, $T$ denotes the length of the target sentence, $X$ is the source sentence, and $\theta$ denotes the model parameters.

Non-autoregressive Language Model removes the internal dependency of target tokens and adopts conditional independent factorization for prediction. Recent non-autoregressive (NAR) language models can be divided into fully NAR and iterative NAR models. The former ones adopt the following objective to maximize the likelihood:

$$\mathcal{L}_{\text{F-NAR}} = \sum_{t}^{T} \log P(y_t | X; \theta), \quad (2)$$

Obviously that the conditional tokens $y_{<t}$ are removed for NAT models. Then the models conduct parallel generation without autoregressive constraints, and the inference speed will be greatly improved. Comparatively, iterative NAR models share a spirit of a mixed autoregressive and non-autoregressive generation. They adopt multiple decoding iterations and keep the autoregressive property in each iteration. They aim to maximize the

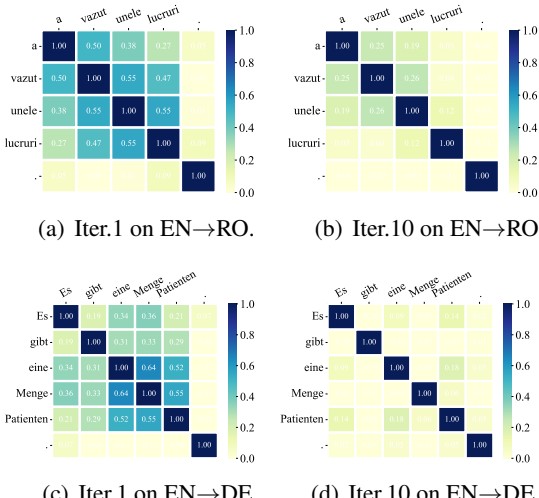

(a) Iter.1 on EN→RO.    (b) Iter.10 on EN→RO.

(c) Iter.1 on EN→DE.    (d) Iter.10 on EN→DE.

Figure 2: Token cosine similarity matrix of CMLM with contrastive learning in iteration 1/10 on raw WMT16 EN→RO and WMT14 EN→DE datasets.

following likelihood:

$$\mathcal{L}_{\text{I-NAR}} = \sum_{t \in Y_{tgt}} \log P(y_t | \hat{Y}, X; \theta), \qquad (3)$$

where the $Y_{tgt}$ denotes the prediction target of current iteration and the $\hat{Y}$ denotes generation result of previous iterations.

## 2.2 Conditional Masked Language Model

Since we adopt the conditional masked language model (CMLM) as the backbone to explore the anisotropic problem for the iterative NAR models, we give a detailed introduction to the training and inference process in CMLM.

**Training Process.** During training, CMLM adopts a masked language modeling task with a uniform masking strategy to decide the masked tokens in the target sentence and learns to predict them. Specifically, given a training pair $(X, Y)$, several tokens in $Y$ will be masked as $Y_{mask}$, and the rest tokens are denoted as $Y_{obs}$. The model then learns to predict the masked tokens $Y_{mask}$ in parallel given $X$ and the unmasked tokens $Y_{obs}$. The training objective of CMLM is to maximize:

$$\mathcal{L}_{\text{CMLM}} = \sum_{y_t \in Y_{mask}} \log P(y_t | Y_{obs}, X; \theta), \qquad (4)$$

where $\theta$ denotes the trainable parameters of CMLM. Besides, as the same as traditional NAR models, an auxiliary task to predict the target length is adopted. CMLM adds a special token [LENGTH] (akin to

| Models | WMT14 | | WMT16 | |
| --- | --- | --- | --- | --- |
| | EN-DE | DE-EN | EN-RO | RO-EN |
| CMLM | 24.92 | 29.59 | 32.84 | 32.44 |
| w/ CL | 25.59 | 29.77 | 32.88 | 32.76 |

Table 1: The results on raw datasets with contrastive learning (CL) for CMLM.

the [cls] token in BERT) into its encoder to predict the target length conditional on the source sentence. **Inference Strategy.** During inference, CMLM adopts the mask-predict decoding algorithm. Specifically, the tokens with low confidence predicted in the previous iteration will be masked, and the model will predict these masked tokens again in the next iteration. Given the target length $L$, the source sentence $X$ and the total decoding iteration $T$, CMLM predicts the entire masked target sentence (i.e., empty $Y_{obs}$) at the first iteration. Then after obtaining the relatively incredible result of the first iteration, the model will choose the specific number of masked tokens with the lowest prediction probability. In an intermediate iteration $t$, the number of the masked tokens $n$ can be calculated as $n = \frac{T-t}{T} * L$. The target sentence with newly masked tokens will be fed into the decoder again and the masked tokens will be predicted conditional on the source sentence $X$ and the unmasked tokens $Y_{obs}$, and the prediction probability of them will be updated. In general, the more iterations, the better the performance of the generation.

## 2.3 Contrastive Learning for CMLM

In this part, we present how to adopt the contrastive learning method for the conditional masked language model (CMLM). Referring to the original paper (Su et al., 2022), we adopt a contrastive objective into the training of CMLM. Specifically, given a training pair $(X, Y)$, the contrastive loss can be calculated as:

$$\mathcal{L}_{\text{sum}} = \sum_{i=1}^{|Y|} \sum_{j=1, j \neq i}^{|Y|} \max\{0, s(h_{y_i}, h_{y_j}) - \rho\}, \quad (5)$$

$$\mathcal{L}_{\text{ave}} = \frac{1}{|Y|(|Y|-1)} \mathcal{L}_{\text{sum}}, \qquad (6)$$

where $|Y|$ is the length of target sentence, $h_{y_i}$ denotes the representation of token $y_i$, $s(h_{y_i}, h_{y_j})$ denotes the cosine similarity between two token representations, and $\rho$ denotes the pre-fined contrastive margin. Intuitively, the contrastive learning

method can punish the close representations of tokens, resulting in a discriminative and isotropic model representation space. The overall training objective for CMLM is defined as:

$$\mathcal{L} = \mathcal{L}_{\text{CMLM}} + \mathcal{L}_{\text{ave}}. \quad (7)$$

Results are shown in Figure 2 (cosine similarity matrix) and Table 1 (BLEU score). Fortunately, the token cosine similarity matrixes become sparse, e.g., the representations of distinct tokens become discriminative significantly and multiple iterations bring further improvements. However, with regard to the BLEU score, there is no consistent improvement for various datasets, e.g., as shown in Table 1, the performances of WMT14 DE→EN and WMT16 EN→RO dataset only have 0.18 and 0.04 BLEU improvement respectively. We analyze the potential reason for this inconsistent improvement and attribute it to the failure to learn the correct representation of the distinct target tokens. More specifically, the anisotropic loss is optimized based on the token representations, so the effectiveness of anisotropic loss is closely related to the model's own ability to learn the token representations, i.e., blindly constraining incorrect token representations by anisotropic loss can't bring substantial improvement for performance although it truly makes their representations more distinguishable. As a result, due to the character that "fail to capture the dependency of target tokens" for NAR models, anisotropic loss just alleviates the traditional anisotropic problem but gets limited improvement on BLEU as shown in Table 1.

It's noticed that "failure to learn representations" is a widely-recognized problem for all NAT models, but not caused by anisotropic loss. The claim denotes that NAR models fail to capture the dependency of target tokens during training (Gu and Kong, 2021a; Xiao et al., 2022; Zhan et al., 2022), i.e., unlike AR models which can learn target-side dependency between one specific token and previous generated tokens, NAR models only depend on source tokens to make predictions due to the conditional independent factorization adopted in training process. As a result, we aim to promote the ability to learn the token representations for CMLM first, then adopt the contrastive learning method.

## 3 Look Neighbors

In this section, we introduce our enhancement method, Look Neighbors, which improves the ability to learn token representations during training. As mentioned in Section 2.3, the failure to learn the token representations well for CMLM prevents the superior effects of adopting contrastive learning. The instability of the representations in NAR models has also been explored in recent NAR papers (Wang et al., 2019; Xie et al., 2022). They propose the regularization method to normalize the token representations. Unlike these regularization methods, we try the enhance the learning of token representations. Specifically, we incorporate the representations of other tokens to make predictions during training, thus improving the ability to learn token representations. As shown in Figure 1, the cosine similarity is denser in adjacent tokens, indicating that CMLM has more difficulty in understanding and distinguishing the adjacent representations of target tokens. As a result, we mainly incorporated the representations of adjacent tokens rather than all other tokens. The most similar exploration is explored in LAVA-NAT (Li et al., 2020). They propose a Look-Around decoding strategy to enhance the inference schedule but without enhancing the training process.

We propose the Look Neighbors, where the model integrates a neighbor-aware self-attention into every decoder layer and concatenates the representations from the last decoder layer of the left and right neighbors to assist in predicting the target tokens. Firstly, we formally introduce our neighbor-aware self-attention as follows:

$$Att_{merge} = \alpha \cdot Att_{self} + (1-\alpha) \cdot Att_{neighbor} \quad (8)$$

$$Att_{neighbor} = softmax(\frac{QK^T}{\sqrt{d}} + M)V \quad (9)$$

$$M_{i,j} = \begin{cases} 0, & j-1 \leq j \leq j+1 \\ -\infty, & otherwise \end{cases} \quad (10)$$

where $\alpha$ is a learnable parameter, $Att_{self}$ is the original decoder self-attention, $Att_{neighbor}$ is our neighbor-aware self-attention, $Att_{merge}$ denotes the final attention we adopt in the experiments, and $M_{ij}$ expresses the position $j$ of the token $i$ in the mask matrix. Next, given the representations of the last decoder layer for target tokens: $H = \{h_1, h_2, ..., h_T\}$, where $h_i$ is the representation of the target token $Y_i$, $T$ is the length of the target sentence. Each representation $h_i$ concatenates the left-neighbor $h_{i-1}$ and right-neighbor $h_{i+1}$, denoted by:

$$h_i' = (W(h_{i-1} \oplus h_{i+1})) + h_i \quad (11)$$

| | Models | $I_{\text{dec}}$ | WMT14 | | WMT16 | | Speedup |
|---|---|---|---|---|---|---|---|
| | | | EN-DE | DE-EN | EN-RO | RO-EN | |
| AT | Transformer* | / | 27.48 | 31.27 | 33.70 | 34.05 | 1.0 × |
| Fully NAT | FT-NAT (Gu et al., 2018) | 1 | 11.79 | 16.27 | - | - | 15.6 × |
| | Flowseq (Ma et al., 2019) | 1 | 23.64 | 28.29 | 32.35 | 32.91 | 1.1 × |
| | Fully-NAT (Gu and Kong, 2021b) | 1 | 23.58 | - | - | - | 16.5 × |
| | AXE-NAT (Marjan et al., 2020) | 1 | 20.40 | 24.90 | - | - | 14.2 × |
| | OAXE-NAT (Du et al., 2021) | 1 | 22.40 | 26.8 | - | - | 15.3 × |
| | GLAT+CTC (Qian et al., 2021) | 1 | 25.02 | 29.14 | - | - | 14.6 × |
| | DSLP+CTC (Huang et al., 2022a) | 1 | 24.81 | 28.33 | - | - | 14.0 × |
| | DA-Transformer (Huang et al., 2022b) | 1 | 26.08 | 30.48 | - | - | 14.0 × |
| | Diff-GLAT (Qian et al., 2022) | 1 | 26.46 | 30.48 | - | - | 14.3 × |
| Iterative NAT | SMART (Ghazvininejad et al., 2020) | 10 | 25.10 | 29.58 | 32.71 | 32.86 | 2.2 × |
| | Disco (Kasai et al., 2020) | 10 | 25.64 | - | - | 32.25 | 3.5 × |
| | Imputer (Saharia et al., 2020) | 8 | 25.00 | - | - | - | 3.9 × |
| | CORR* (Huang et al., 2022c) | 10 | 26.01 | 30.55 | 33.71 | 33.27 | 2.1 × |
| | CMLMC (Huang et al., 2022c) | 10 | 26.40 | 30.92 | 34.14 | 34.13 | 1.7 × |
| Ours | CMLM* (Ghazvininejad et al., 2019) | 10 | 24.92 | 29.59 | 32.84 | 32.44 | 2.2 × |
| | CMLM w/ Ours | 10 | **26.68** | **30.50** | **34.01** | **33.83** | 1.7 × |

Table 2: Results on 4 WMT machine translation tasks. * denotes our implementations.

where $W$ is a learnable matrix, $\oplus$ denotes the matrix concatenation. Then the decoder representations will be $H' = \{h'_1, h'_2, ..., h'_T\}$, and the model predict the tokens based on $H'$. Obviously, the Look Neighbors strategy makes each representation to be aware of the information about its neighbors, thus improving the learning of token representations during training.

## 4 Experiment

### 4.1 Datasets

We verify the effectiveness of our method by conducting experiments on multiple public translation datasets: WMT14 EN↔DE, WMT16 EN↔RO. The WMT En↔Ro and En↔De datasets contain $0.6M/1.9K/1.9K$ and $4.5M/39K/3K$ of training/valid/test sentence pairs, respectively.

### 4.2 Experimental Settings

We conduct all experiments using the Fairseq library (Ott et al., 2019). We adopt the same model structure for a fair comparison with previous work and the CMLM baseline, i.e., the standard Transformer$_{base}$ configuration containing 6 layers per stack, 8 attention heads per layer, 512 model dimensions, and 2048 hidden dimensions. We adopt the Adam optimizer (Kingma and Ba, 2014) $\beta$ with (0.9, 0.98), set weight decay to 0.01

and label smoothing to 0.1 for all the experiments. We train the models with batches of 64k tokens for all the datasets. We set the dropout rate as 0.2 to EN↔DE datasets and 0.3 to EN↔RO datasets. The other hyper-parameters are adopted following CMLMC (Huang et al., 2022c), e.g., the learning rate warms up to 7e-4 in 40k for WMT14 EN↔DE datasets, then gradually decays with an inverse square schedule for 300k update steps in total. The corresponding settings for WMT16 EN↔RO datasets are 5e-4/15k/100k. Besides, notice that the pre-fined contrastive margin $\rho$ is set to 0.4, which has proven to achieve the best performance in 4.4. During inference, we average the 5 best checkpoints based on validation BLEU scores as our final model, and the length beam is set to 5.

### 4.3 Results

Following previous works, we evaluate the performance with BLEU (Papineni et al., 2002). Results are presented in Table 2, we compare our methods with the traditional CMLM, the strong autoregressive baseline: vanilla Transformer, and other NAT methods including related methods (NAT-REG, Fully-NAT), recent competitive Fully NAT models (GLAT, DSLP, DAD and DA-Transformer), and several CMLM-based iterative NAT models (SMART, Disco, Imputer, MvSR-NAT, CCMLM

| Models | WMT14 | | WMT16 | |
|---|---|---|---|---|
| | EN-DE | DE-EN | EN-RO | RO-EN |
| CMLM | 24.92 | 29.59 | 32.84 | 32.44 |
| w/ LN | 25.41 | 30.36 | 33.43 | 33.02 |

Table 3: The effect of LN for CMLM on WMT datasets.

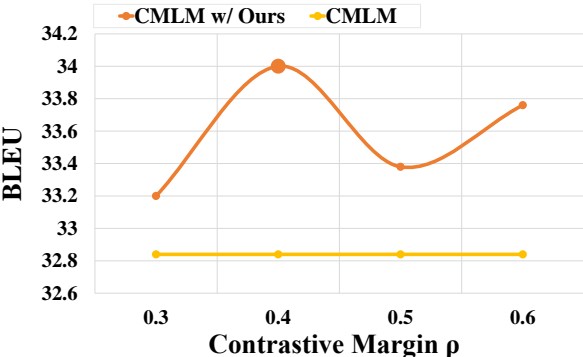

Figure 3: The exploration of the best pre-fined contrastive margin $\rho$.

and CMLMC). Firstly, our methods gain improvements on all datasets compared with traditional CMLM, and mitigate the gap between AT and NAT models for about 1.0 BLEU score. Compared with other related NAT methods, our proposed methods outperform most of them, except for CMLMC. Surprisingly, our methods achieve 26.68 on WMT14 EN→DE, and set the new SoTA performance. Besides, compared with CMLM w/ CL (shown in Table 1), CMLM with both methods achieves significant improvements. It verifies our hypothesis that while better capturing the dependency of target tokens for NAR models, the anisotropic loss can perform more effectively to improve the performance while alleviating the anisotropic problem. We also provide the case study in Appendix A to further understand the effectiveness of our method.

### 4.4 Ablation Study

**Effect of Look Neighbors** In order to evaluate the effectiveness of Look Neighbors, we conduct the experiments by comparing CMLM with/without Look Neighbors (LN). As shown in Table 3, it's evident that CMLM w/ LN achieves consistent improvements on BLEU score (25.41 vs. 24.92, 33.43 vs. 32.84), which certifies that the Look Neighbors strategy is able to capture the dependency of the adjacent tokens effectively and assist in generating better translation results.

| Models | WMT14 EN → DE | WMT16 EN → RO |
|---|---|---|
| CMLM | 27.21 | 33.36 |
| w/ Ours | 27.71 | 34.33 |

Table 4: The performance of CMLM w/o and w/ our methods on WMT distilled data.

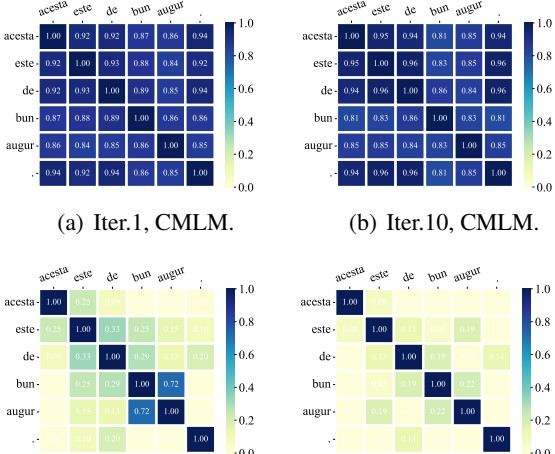

(a) Iter.1, CMLM.  (b) Iter.10, CMLM.

(c) Iter.1, CMLM w/ Ours.  (d) Iter.10, CMLM w/ Ours.

Figure 4: Token cosine similarity matrix of CMLM w/o and w/ our methods in iteration 1/10 on distill WMT16 EN→RO dataset.

**Best Contrastive Margin** As mentioned in Section 2.3, there is a pre-fined contrastive margin $\rho$ to control the strength of contrastive learning. If the $\rho$ is relatively big, the effect of contrastive learning is small. However, once the $\rho$ becomes 1.0, the contrastive loss has no effect on training. We conduct experiments on WMT16 EN→RO dataset and set the margin from 0.3 to 0.6. As shown in Figure 3, the performance all outperforms the vanilla CMLM, indicating that it is not hard to apply our methods. The best value is 0.4.

## 5 Analysis

### 5.1 Performance on Distilled Data

Knowledge distillation is a widely-used method in various NAR models (Hinton et al., 2015; Zhou et al., 2019). Following the previous works (Gu et al., 2018; Huang et al., 2022c), we first adopt AR models trained with original data to generate the results. Then the generated outputs are adopted as distilled data and serve as the training data of NAR models. We adopt Transformer$_{base/big}$ as the teacher model for WMT EN↔RO and EN↔DE, respectively. Compared with original training data, distilled data is more explicit and simpler, and can

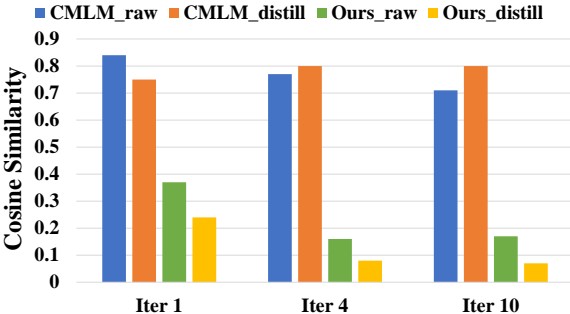

Figure 5: Comparison of token similarity distribution between CMLM w/o and w/ our methods.

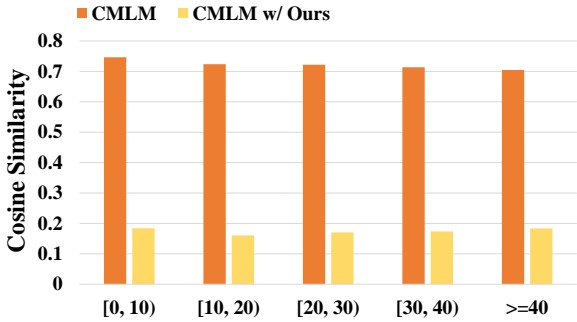

Figure 6: Comparison of token similarity distribution between CMLM w/o and w/ our methods with different source lengths.

| Models | Iter | WMT14 EN → DE | | | WMT16 EN → RO | | |
|--------|------|------|------|-------|------|------|-------|
|        |      | BLEU | Reps | Comet | BLEU | Reps | Comet |
| CMLM   | 4    | 22.50 | 2.00% | -0.09 | 31.47 | 0.90% | 0.21 |
|        | 10   | 24.92 | 0.31% | 0.08  | 32.84 | 0.24% | 0.28 |
| w/ Ours | 4   | 24.07 | 1.18% | -0.01 | 32.79 | 0.72% | 0.26 |
|        | 10   | 26.68 | 0.17% | 0.12  | 34.01 | 0.13% | 0.31 |

Table 5: BLEU score, the token repetition ratio and Comet score with different iterations on the raw WMT14 EN → DE and WMT16 EN → RO datasets.

help alleviate the well-known multi-modality problem in NAR models. We also evaluate our methods on distilled data and present the results in Table 4. Our method can achieve about 0.5/1.0 BLEU score improvements on WMT EN→DE and EN→RO, respectively. Besides, we also compare the token cosine similarity with/without our methods on distilled data. We plot the results in Figure 4, and we can find that training with distilled data still suffer from the anisotropic problem, and our methods also bring significant benefits in alleviating this problem.

## 5.2 Quantitative Comparison

**Cosine Similarity.** Many previous works have pointed out the positive effects of iterative decoding and knowledge distillation (Ding et al., 2020; Gu and Kong, 2021b; Xiao et al., 2022) for NAR models. In order to better understand the effects of them for alleviating the anisotropic problem, we conduct a quantitative comparison of the token cosine similarity between different iterations and knowledge distillation. As we only compare the cosine similarity with one specific example in the previous contexts, e.g., Figure 1 and Figure 2, we aim to explore the effects on the full test sets and compute the average token cosine similarity scores of WMT16 EN → RO dataset and plot them in Figure 5. We find that decoding with more itera-

tions and adopting knowledge distillation does not alleviate the anisotropic problem effectively. Besides, adopting our methods presents significant effects in reducing the token cosine similarity scores. There are no conflicts in our methods with iterative decoding or knowledge distillation, i.e., adopting knowledge distillation with our methods together further reduces the token cosine similarity scores, and the token cosine similarity scores trained with our methods will be lower as more iterations are adopted.

**Token Repetition and Comet.** Repetition problem is common in NAR models, we view the anisotropic problem as one of the reasons to cause token repetition. Since the anisotropic problem can be alleviated well with our methods, we wonder if the repetition rate reduces. We compare our methods with vanilla CMLM of different iterations and compute the corresponding repetition ratios. Besides, we also report Comet (Rei et al., 2020) to make a more complete evaluation. The results on raw WMT EN→DE and EN→RO datasets are presented in Table 5. As we can see, our method achieves a consistent decrease in the repetition ratio with different iterations. Besides, the BLEU score and Comet also outperform the vanilla CMLM in all conditions.

**Different Source Lengths.** We also explore the effects of our methods with different sentence lengths. Specifically, we divide the source sentences of WMT16 EN→RO test set into 5 intervals by the length after BPE operation, then compare the average token representation similarity of iteration 10 with/without our methods. Results are shown in Figure 6. It seems the length has a small impact on token representation similarity, and our

| Models | WMT16 EN → RO | WMT16 RO → EN |
|---|---|---|
| CMLM | 32.84 | 32.44 |
| CORR | 33.71 | 33.27 |
| CORR w/ Ours | **34.15** | 34.02 |

Table 6: The performance of CMLM combined with CORR and CORR w/ our method.

methods alleviate the anisotropic problem greatly with all conditions of sentence length, e.g., the token representation similarity is all about 0.7 with different source sentence lengths and reduced to around 0.15 with our methods. Although the token representation similarity seems to increase for longer sentences after adopting our methods, the final result is still less than 0.2, and this small increase can be ignored.

### 5.3 More Explorations

As shown in Table 2, the performance of our methods is comparable with CMLMC (Huang et al., 2022c), which introduces a self-correction mechanism based on CMLM. We also explore the potential of adopting our methods to combine with this mechanism. We only adopt the self-correction mechanism by ourselves as the same as CMLMC based on CMLM (CORR), and then adopt our methods (contrastive learning and look neighbors) to it. Results on two WMT datasets are shown in Table 6, we can find that the self-correction mechanism truly improves the performance of vanilla CMLM, and training with our methods further boosts the performance. Moreover, we also plot the token cosine similarity matrix of CORR with/without our methods in Figure 7. Results show that adopting the self-correction rather than just the mask-predict algorithm in vanilla CMLM does not help alleviate the anisotropic problem, and our methods also bring benefits with a self-correction mechanism. This also presents the potential of our methods to be applied to other iterative NAR models.

### 6 Related Work

The iterative NAR models are proposed to achieve the trade-off between inference speedup and generation quality. Lee et al. first propose the iterative model, where the noised target sentences are refined in multiple iterations. Later, text-editing methods (Stern et al., 2019; Gu et al., 2019; Lu and Peng, 2022) are also introduced. They generate the target sentences by adopting the auxiliary insertion and deletion operations in each iter-

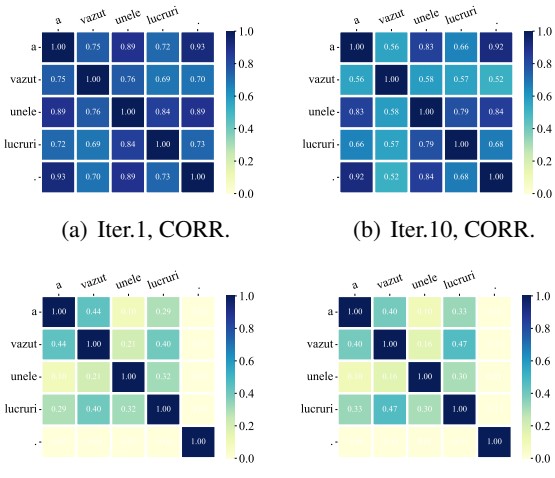

(a) Iter.1, CORR.  (b) Iter.10, CORR.

(c) Iter.1, CORR w/ Ours.  (d) Iter.10, CORR w/ Ours.

Figure 7: Token cosine similarity matrix of CORR w/o and w/ our methods in iteration 1/10 on raw WMT16 EN→RO dataset.

ation. Besides, motivated by BERT (Kenton and Toutanova, 2019), Ghazvininejad et al. propose the conditional masked language model (CMLM) which adopts the uniform masking strategy during training and a mask-predict strategy for inference. Due to its promising performance, CMLM has become one of the most competitive and widely-used iterative NAR models, and many advanced enhancement strategies from different perspectives have been proposed based on CMLM in recent years, e.g., optimizing the inference strategy (Kasai et al., 2020; Geng et al., 2021), optimizing the masking strategy (Guo et al., 2020; Xiao et al., 2023), benefiting from the AT counterpart (Hao et al., 2021; Xiaobo Liang and Zhang, 2022), training with better criterion (Marjan et al., 2020; Du et al., 2021) and introducing correction mechanism (Ghazvininejad et al., 2020; Huang et al., 2022c). Recently, motivated by the application of diffusion models (Ho et al., 2020), Savinov et al. proposed a step-unrolled denoising autoencoder that adopts a denoising operation in each iteration. Besides, Wang et al. adapt the cross-lingual pre-training model (XLMR) into NAT models and further improve the performance with iterative refinements.

The token representations of NAR models have also attracted much attention once the birth of NAR models. Wang et al. normalize the learning of the decoder representations by introducing similarity and reconstruction regularizations. Xie et al. propose a consistency training method to regularize the

representations of the same tokens in the different masked target sentences, thus improving the stability of representing the tokens. Besides, the contrastive learning method has been explored in (Su et al., 2022) to pull the positive token pairs together and push negative pairs away. They optimize the similarity of several different representations of the same token to learn more informative and robust representations. Unlike this contrastive method, we try to make the representations of tokens in one sentence more discriminative, thus alleviating the problem of anisotropic representations. The anisotropic problem is proposed in (Su et al., 2022), they focus on the AR models and try to alleviate the degeneration problem with the explorations of the anisotropic problem. In this paper, we focus on the iterative NAR model CMLM and try to explore more potential of iterative models.

## 7 Conclusion

In this paper, we first verify that the strong and representative iterative NAR model, CMLM, suffers from the anisotropic problem by analyzing the changes in the cosine similarity of token representations. To alleviate the anisotropic problem, we introduce the contrastive learning method to make the token representations discriminative and further propose a Look Neighbors strategy to enhance token representation learning during training. Extensive experiments on several widely-used translation datasets indicate that our proposed methods have a great effect on alleviating the anisotropic problem and can consistently improve the performance of iterative NAR models. We hope that our explorations can give new insights into the iterative NAR models and promote their development.

## 8 Limitations

In this section, we present several limitations of our proposed methods and this paper. Firstly, we only take the conditional masked language model (CMLM) as a backbone to evaluate the effectiveness of our methods, more iterative models may also need exploration, e.g., the recent diffusion models. Besides, since the repetition problem is quite serious in fully NAR models, whether the anisotropic problem exists in fully NAR models, we leave this as a future work. Secondly, since singly adopting contrastive learning brings few benefits on the performance as mentioned in Section 2.3, we further propose the Look Neighbors

to enhance the learning of token representations. As a result, we wonder whether there exist other enhancing methods that can better combine with contrastive learning.

## 9 Acknowledge

This work is supported by the National Science Foundation of China (NSFC No. 62206194) and the Natural Science Foundation of Jiangsu Province (Grant No. BK20220488).

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

| | Example 1 | Example 2 |
|---|---|---|
| Source | If an affected person has waited for a sufficiently long period of time at a red light , and provided the crossing is clear , they can drive on , explained Stuttgart @-@ based lawyer Ralf Becker in " Motorrad " ( Motorcycle ) magazine . | Aside from honouring Hugo , the 31st Autumn Festival progressed as usual : Alongside the Betra Male Voice Choir , the Salzstetten Choral Club , the Baisingen Choral Division and the Local Music Society , the Vollmaringen singers delivered a colourful blend of different choral and song styles , which entertained the 400 or so visitors . |
| Reference | Hat ein Betroffener lange genug an einer roten Ampel gewartet und die Kreuzung ist frei , kann er weiterfahren , erklärt der Stuttgarter Rechtsanwalt Ralf Becker in der Zeitschrift " Motorrad " . | Neben der Ehrung verlief das 31. Herbstfest in gewohnten Bahnen : Mit dem MGV Betra , dem Liederkranz Salzstetten , der Sängerabteilung Baisingen und dem örtlichen Musikverein hatten die Vollmaringer eine bunte Mischung verschiedener Chöre und Gesangsstile geladen , welche die rund 400 Besucher unterhielten . |
| CMLM | Wenn eine betroffene Person eine ausreichend lange Zeit im roten Licht gewartet hat und sofern der Übergang klar ist , kann sie weiterfahren , erklärte der Stuttgarter Rechtsanwalt Ralf Becker im " Motorrad rad " ( Motorcycle ) . | Neben der Ehrung von Hugo entwickelte sich das 31. Herbstfest wie üblich : Neben dem Betra Male Voice Chor , dem Chorclub SalzSalzstetten , der Chorabteilung Baisingen ingen und der lokalen Musikgesellschaft gaben die Vollmaringen ingen ingen ingen eine farbenfrohe Mischung aus verschiedenen Chor- und Songstilen , , an die etwa 400 Besucher untertierten . |
| CMLM w/Ours | Wenn der Betroffene schon eine ausreichend lange Zeit an einem roten Licht gewartet hat und sofern der übergang klar ist , kann er weiterfahren , erklärte der Stuttgarter Rechtsanwalt Ralf Becker in der Zeitschrift " Motorrad " . | Neben der Ehren von Hugo schreitet das 31. Herbstfest wie üblich voran : Neben dem Betra Male Voice Chor , dem Salzstetten Chor Club , der Chorabteilung Baisingen und der Local Music Society lieferten die Vollmaringer SänSänger eine farbenfrohe Mischung aus verschiedenen Chor- und Liongstilen , die etwa 400 Besucher begleiteten . |

Table 7: Translation examples of CMLM w/ and w/o our methods on WMT14 EN→DE test set.

## A Case Study

We provide two case analyses that experimented (shown in Table 7) on WMT14 EN-DE dataset to understand the effectiveness of our method better.

- In Example 1, we can find that the result of CMLM obtains the words "Motorrad rad" which is a grammatical error and the translation of the word 'magazine' is missed. In contrast, our methods enhance the dependency of token representations to alleviate the above problems.

- In Example 2, the words 'ingen' and ',' are repeated several times in the result of CMLM which is caused by the anisotropy problem. Our methods are effective in alleviating this problem and provide more fluent translation.