# OpenReview forum: "Isotropy-Enhanced Conditional Masked Language Models"
_EMNLP/2023/Conference — EMNLP 2023 Findings_

### Official Review · Reviewer_Nvxc · 2023-08-05

**Soundness:** 3

**Excitement:**

3: Ambivalent: It has merits (e.g., it reports state-of-the-art results, the idea is nice), but there are key weaknesses (e.g., it describes incremental work), and it can significantly benefit from another round of revision. However, I won't object to accepting it if my co-reviewers champion it.

**Paper Topic And Main Contributions:**

This paper addresses the problem of anisotropic token representations in iterative non-autoregressive neural machine translation models like CMLM. The main contributions are:

- Analyzing and confirming the anisotropic problem (similar token representations) exists in iterative NAR models like CMLM.
- Proposing two methods to alleviate this problem: (i)Using contrastive learning to make token representations more discriminative. (ii) Introducing a "Look Neighbors" strategy to enhance learning of adjacent token representations.
- Quantitatively and qualitatively showing the proposed methods can reduce token similarity and improve performance across multiple WMT datasets.
- Demonstrating the proposed methods are compatible and complementary to other recent improvements for iterative NAR like knowledge distillation and self-correction.

In summary, this paper identifies and provides solutions for the anisotropic token representation problem in iterative non-autoregressive translation models. The analysis and proposed methods help improve performance and point towards future work on better token representations for NAR.

**Reasons To Accept:**

- Addresses an important problem (anisotropy) in an impactful area (iterative NAR MT). Improving NAR is an active area of research.
- Provides solid empirical analysis of the problem using visualizations and metrics.
- Proposes two simple but effective solutions that consistently improve performance across multiple datasets. And ablations validate the contribution of them.
- Compatibility with other recent advances like knowledge distillation and self-correction is demonstrated.

**Reasons To Reject:**

- The performance gains are decent but not dramatic compared to prior iterative NAR methods.
- The analysis is mainly limited to one model (CMLM). More NAR models could be explored. The paper does seem to be missing a discussion of some important related work on token representations in non-autoregressive models: (i) NAT-REG (Wang et al. 2019) - Proposed similarity and reconstruction regularizations to normalize decoder token representations in NAR models. This seems highly relevant to the goal of reducing anisotropy. (ii) LAVA-NAT (Li et al. 2020) - Uses a "Look-Around" strategy at inference time to incorporate context representations. Related to the proposed "Look Neighbors". Comparing to NAT-REG may help better motivate the need for contrastive learning and neighbor-aware training. The connection to LAVA-NAT's look-around decoding could also be clarified.
- There could be more exploration into why exactly anisotropy arises and how it impacts generation.
- Missing strong baselines.  Diff-GLAT (https://arxiv.org/abs/2212.10240)

**Reproducibility:**

4: Could mostly reproduce the results, but there may be some variation because of sample variance or minor variations in their interpretation of the protocol or method.

**Reviewer Confidence:**

5: Positive that my evaluation is correct. I read the paper very carefully and I am very familiar with related work.

---

> ### Author Rebuttal · Authors · 2023-08-29
>
> Thanks for your suggestions and comments, we give more explanations for your concerns and questions.
>
> **For your concern 1**
> - Our methods gain the average of **1.3** BLEU improvements compared with traditional CMLM on all datasets, and **our methods achieve 26.68 on WMT14 EN->DE which set the new SoTA performance**. What's more, it's compatible to apply our methods on the other iterative NAR models such as CORR (Table 6) to further improve the performance.
>
> **For your concern 2**
> - Thanks for your advice. CMLM is a classic and representive model that many related work focus on, so we pay attention to explore it. For two related work you propose:
> (1) Actually, we have adopted the similarity and reconstruction regularizations methods proposed in NAT-REG on CMLM and found that it couldn't alleviate the anisotropic problem.
> (2) We present the difference between our Look Neighbors and LAVA-NAT's look-around from line 267 to 271 in our paper. The look-around is a decoding strategy to enhance the inference schedule without enhancing the training process, our method works during training and inference.
>
> **For your concern 3**
> - Anisotropy problem is common on Transformer-based models, about why it arises and how it impacts generation could be found in (Kawin Ethayarajh 2019). Our paper mainly pay attention to alleviate this problem and improve the performance.
>
> **For your concern 4**
> - Thanks for your reminder, we will add it in the next version. From Diff-GLAT, we can see that our methods outperform diff-GLAT (CTC) on WMT14 EN-DE. Combining with strong baseline DAT making diff-GLAT improve greatly, we will adopt our methods on DAT for fair comparison.
>
> **Reference:**
> - Kawin Ethayarajh. EMNLP, 2019. How Contextual are Contextualized Word Representations? Comparing the Geometry of BERT, ELMo, and GPT-2 Embeddings.

---

### Official Review · Reviewer_JroP · 2023-08-09

**Typos Grammar Style And Presentation Improvements:** n/a
**Soundness:** 3

**Excitement:**

3: Ambivalent: It has merits (e.g., it reports state-of-the-art results, the idea is nice), but there are key weaknesses (e.g., it describes incremental work), and it can significantly benefit from another round of revision. However, I won't object to accepting it if my co-reviewers champion it.

**Missing References:**

n/a

**Paper Topic And Main Contributions:**

This paper tackles non-autoregressive text generation, particularly in the applications of machine translation.  Inspired by recent study on the anisotropic property of language models (the hidden representations at different positions are too similar and indiscriminate), the authors propose to encourage dissimilarity by a contrastive loss. they further propose a Look Neighbors mechanism. Experiments show that  the proposed method obtain better results than vanilla NAR models and many existing baselines.

**Questions For The Authors:**

n/a

**Reasons To Accept:**

1. The authors presents many insightful experimental results (thought I find many of them may not well contribute to the main theme of this paper).

**Reasons To Reject:**

1. I find the motivation for the Look Neighbors (LN) mechanism vague. Why do you mean by "failure to learn the correct representation"? This almost conveys no information because one can always says the representation is not correct when the results are not good.
2. What are the connections between contrastive loss (CL) and LN? Can LN be used without CL? (ablation study is needed) I find that the combinations of these two makes the paper even confusing. Because contrastive loss is meant to make the representations at different positions different. On the other hand, LN intends to make the representations focus on near neighbors rather than distant neighbors. I think it may make near neighborhood even more similar.
3. Comparing Table 1 and Table 3, WMT14 EN → DE: 24.92 ->25.59 with CL and WMT14 EN → DE: 24.92 ->25.41 with LK. It seems that the LK mechanism has negative impact on the performance compared to simple CL. This makes the effect of LK suspicious.

Overall, I think the motivation for the proposed LK mechanism has been not well-justified either theoretically or empirically.

**Reproducibility:**

3: Could reproduce the results with some difficulty. The settings of parameters are underspecified or subjectively determined; the training/evaluation data are not widely available.

**Reviewer Confidence:**

4: Quite sure. I tried to check the important points carefully. It's unlikely, though conceivable, that I missed something that should affect my ratings.

---

> ### Author Rebuttal · Authors · 2023-08-29
>
> Thanks for your suggestions and comments, we give more explanations for your concerns and questions.
>
> **For your concern 1**
> - We are sorry to make the description vague in the paper. Now, we make it clearer here.
>   - **The claim that "failure to learn representations" is a widely-recognized problem for all NAT models, but not caused by anisotropic loss.** The claim denotes that NAR models **fail to capture the dependency of target tokens** during training (Gu and Kong 2021; Xiao et al 2022; Zhan et al 2022), i.e., unlike AR models (e.g., vanilla Transformer) which can learn target-side dependency between one specific token and previous generated tokens, NAR models only depend on source tokens to make predictions due to the conditional independent factorization adopted in training process. The motivation for the Look Neighbors could be found in response to concern 2.
>   - **Anisotropic problem is another common problem for natural language models (Kawin Ethayarajh 2019, Su et al 2022), which has no direct connection with the problem "failure to learn representations" of NAR models.**
>
> **For your concern 2**
> - We further explain the connections between CL and LN here:
> **As we know, anisotropic loss is optimized based on the token representations, so the effectiveness of anisotropic loss is closely related to the model's own ability to learn the token representations**, i.e., Easy to understand, blindly constraining incorrect token representations by anisotropic loss can't bring substantial improvement for performance although it truly make their representations more distinguishable. As a result, due to the character that "fail to capture the dependency of target tokens" for NAR models, anisotropic loss just only alleviates the traditional anisotropic problem but gets the limited improvement on BLEU as shown in Table 1. Therefore, we first propose the Look Neighbors to enhance the learning of representation dependency. While better capturing the dependency of target tokens for NAR models, anisotropic loss can perform more effectively to improve the performance while alleviating the anisotropic problem. You can refer the results of table shown in response to concern 3 to further understand the connections.
> - For your another concerns, we make more detailed presentations. Firstly, **LN could be used without CL**, we provide the ablation study in Section 4.4 'Effect of Look Neighbors'. Besides, we compute the cosine similarity with CMLM w/ LN, and find that **it is consistent with CMLM and doesn't make near neighborhood more similar.**
>
> **For your concern 3**
> - Now, we provide the results of CMLM w/ LN, w/ CL and w/ both. As shown in table, simply adoting LN can consistently improve the performance on all datastes. Besides, CL and LN are independent not contradictory. They are independent according to two different well-known problems. LN can make CL more effective in improving the performance while alleviating the anisotropic problem.
>
> |              | WMT14 EN-DE | WMT14 DE-EN | WMT16 EN-RO | WMT16 RO-EN |
> | :----------: | :---------: | :---------: | :---------: | :---------: |
> |     CMLM     |    24.92    |    29.59    |    32.84    |    32.44    |
> |  CMLM w/ CL  |    25.59    |    29.77    |    32.88    |    32.76    |
> |  CMLM w/ LN  |    25.41    |    30.36    |    33.43    |    33.02    |
> | CMLM w/ both |    26.68    |    30.50    |    34.01    |    33.83    |
>
> **Reference:**
>
> - Gu J. and Kong X. ACL, 2021. Fully Non-autoregressive Neural Machine Translation: Tricks of the Trade.
> - Xiao, Y.; Wu, L.; Guo, J.; Li, J.; Zhang, M.; Qin, T.; and Liu, T.-y. TPAMI, 2022. A Survey on Non-Autoregressive Generation for Neural Machine Translation and Beyond.
> - Zhan, J.; Chen, Q.; Chen, B.; Wang, W.; Bai, Y.; and Gao, Y. Arxiv 2022. Non-autoregressive Translation with Dependency-Aware Decoder.

---

### Official Review · Reviewer_azMj · 2023-08-11

**Soundness:** 3

**Excitement:**

4: Strong: This paper deepens the understanding of some phenomenon or lowers the barriers to an existing research direction.

**Paper Topic And Main Contributions:**

This paper is an exploration on the effects of anisotropy in conditional masked (CMLM) non-autoregressive machine translation models. The paper reports experiment results on CMLM models trained with regularization penalizing anisotropy. It also propose a "Look Neighbor" method promoting neighboring tokens' information when decoding the current position. The paper reports limited positive experimental results for both methods.

**Reasons To Accept:**

* The paper investigates a very interesting research question and perform some thoughtful experiments to answer it to limited extents.
  * Specifically, the experiments on applying anisotropy regularization is interesting, but the data reported is quite limited.
* The paper also proposes the Look Neighbor method which seems to have a positive impact on CMLM, though further investigation is warranted.
* The introduction (up to line 85) is well-written and thought-provoking.

**Reasons To Reject:**

* The reasoning is often times disconnected at various key moments in the paper. In paragraph from line 96 to line 109, as well as on lines 230-233, the authors abruptly transitions from anisotropic loss not helping to model's "failure to learn representations". This appears to be a key point in the paper that's repeated several times. However, there are several problems with it:
  * "Failure to learn the correct representation" is too vague -- what *specific* failures is in question here? Too many things can be blamed under the umbrella "not learning the correct representation".
  * There is almost no logical connection between anisotropic loss not helping to the conclusion that authors are trying to draw.
  * There isn't sufficient data to support authors' claim that anisotropic loss is not helping. In table 1, it appears that the loss either improves or doesn't degrade CMLM's performance, and moreover, this table's data is inconsistent with the rest of the paper, where both directions for both language pairs are reported. Why are the DE->EN and RO->EN directions missing?
* The Look Neighbors method isn't well motivated. It seems very disconnected with the "isotropy" topic of the present paper.
* There are some confusions with the presentation. Since the isotropy loss (called "CL" in the paper) and Look Neighbors (LN) are two distinct methods, what does "ours" mean in Table 2? Is it CL+LN or just LN? And what about the ablation studies in Section  5? Are those models with or without CL? And what about Figure 3? Are the models in Figure 3 trained with only CL or CL+LN? Moreover, the numbers in Figure 3 seem to be very close to each other. It seems doubtful that we can draw any statistically significant conclusions from this graph (the same goes for the second column in Table 1).

Overall, I think this is promising research direction and paper would be an interesting read if the above problems are fixed. The author might also consider splitting CL and LN into two separate papers because they seem like two different topics. However, I wouldn't recommend acceptance of the paper at the current form.

**Reproducibility:**

2: Would be hard pressed to reproduce the results. The contribution depends on data that are simply not available outside the author's institution or consortium; not enough details are provided.

**Reviewer Confidence:**

5: Positive that my evaluation is correct. I read the paper very carefully and I am very familiar with related work.

**Typos Grammar Style And Presentation Improvements:**

* Line 141-142: this sentence is ill-formed.
* Eq. 6 (line 209): From context, I think the denominator should be $|Y|(|Y|-1)$ instead of $|Y||Y|-1$.

---

> ### Author Rebuttal · Authors · 2023-08-29
>
> Thanks for your suggestions and comments, we give more explanations for your concerns and questions.
>
> **For your concern 1**
> - We are sorry to make the description vague in the paper, and may cause you to misunderstand it. Now, we make it clearer here.
>   - **The claim that "failure to learn representations" is a widely-recognized problem for all NAT models, but not caused by anisotropic loss.** The claim denotes that NAR models **fail to capture the dependency of target tokens** during training (Gu and Kong 2021; Xiao et al 2022; Zhan et al 2022), i.e., unlike AR models (e.g., vanilla Transformer) which can learn target-side dependency between one specific token and previous generated tokens, NAR models only depend on source tokens to make predictions due to the conditional independent factorization adopted in training process.
>   - **Anisotropic problem is another common problem for natural language models (Kawin Ethayarajh 2019, Su et al 2022), which has no direct connection with the problem "failure to learn representations" of NAR models.**
>   Notice we just claim the "inconsistent improvement" in our paper but not that "anisotropic loss is not helping" as you said. As shown in Figure 2, adopting anisotropic loss is truly an effective way to mitigate anisotropic problem, but not effective enough to improve the performance. This is actually the main claim in our paper, we further explain it now:
>   - **As we know, anisotropic loss is optimized based on the token representations, so the effectiveness of anisotropic loss is closely related to the model's own ability to learn the token representations,** i.e., Easy to understand, blindly constraining incorrect token representations by anisotropic loss can't bring substantial improvement for performance although it truly make their representations more distinguishable. As a result, due to the character that "fail to capture the dependency of target tokens" for NAR models, anisotropic loss just only alleviates the traditional anisotropic problem but gets the limited improvement on BLEU as shown in Table 1. (we also conduct experiments on other directions as you mentioned w/o and w/ CL, WMT DE->EN: 29.59 vs. 29.77, and RO-EN 32.44 vs. 32.76, we will supply them in the paper.) **Therefore, we propose the Look Neighbors to enhance the learning of representation dependency.**
>
> **For your concern 2**
> - **We believe that we have present the difference of two problems clearly mentioned in the response to concern 1. The Look Neighbors is just to enhance the learning of token representations, but have no direct connection for the anisotropic problem.**
> While better capturing the dependency of target tokens for NAR models, anisotropic loss can perform more effectively to improve the performance while alleviating the anisotropic problem. Besides, we explain why we adopt Look Neighbors to enhance the learning of NAR models. Compared with other methods, Look Neighbors is simple yet effective, and why only Neigbours, as shown in previous works, adjacent tokens are more important for the learning in NAR models (Liang et al 2023).
>
> **For your concern 3**
> - **"Ours" means CL + LN two methods in all of the Tables and Figures.**
> The experiment in Figure 3 is to explore the best pre-fined contrastive margin ρ. In figure 3, lower part line denotes CMLM without ours, so the performance is all the same. The other line denotes the results of our method (CL + LN) with different ρ (0.3, 0.4, 0.5, 0.6), the curve reaches its peak when ρ is 0.4, and notice that compared with 0.3, it has 0.8 BLEU improvement, we do not think this is close to each other. Besides, in Table 1, the result is closed in the second column. Notice this is why we claim that "inconsistent improvement" of anisotropic loss, and after we adopt Look Neighbours method to enhance the learning of NAR models, anisotropic loss can perform quite well to improve the performance, as shown in Table 2 (24.92 vs 26.68, 29.59 vs. 30.50, 32.84 vs. 34.01, 32.44 vs. 33.83).
>
>
> **The responses to 'Typos Grammar Style And Presentation Improvements'**
>
> - Thanks for your reminder, we have fixed them.
>
> **Reference:**
>
> - Gu J. and Kong X. ACL, 2021. Fully Non-autoregressive Neural Machine Translation: Tricks of the Trade.
> - Xiao, Y.; Wu, L.; Guo, J.; Li, J.; Zhang, M.; Qin, T.; and Liu, T.-y. TPAMI, 2022. A Survey on Non-Autoregressive Generation for Neural Machine Translation and Beyond.
> - Zhan, J.; Chen, Q.; Chen, B.; Wang, W.; Bai, Y.; and Gao, Y. Arxiv 2022. Non-autoregressive Translation with Dependency-Aware Decoder.
> - Kawin Ethayarajh. EMNLP, 2019. How Contextual are Contextualized Word Representations? Comparing the Geometry of BERT, ELMo, and GPT-2 Embeddings.
> - Yixuan Su, Tian Lan, Yan Wang, Dani Yogatama, Ling peng Kong, and Nigel Collier. NeurIPS, 2022. A contrastive framework for neural text generation.
> - Xiaobo Liang, Zecheng Tang, Juntao Li, and Min Zhang, ACL, 2023. Open-ended Long Text Generation via Masked Language Modeling.

---

### Official Review · Reviewer_ptyC · 2023-08-12

**Soundness:** 3

**Excitement:**

3: Ambivalent: It has merits (e.g., it reports state-of-the-art results, the idea is nice), but there are key weaknesses (e.g., it describes incremental work), and it can significantly benefit from another round of revision. However, I won't object to accepting it if my co-reviewers champion it.

**Paper Topic And Main Contributions:**

The authors delved deeper into the problems faced by iterative non-autoregressive models, specifically the issue of similar and indiscriminative representations of predicted target tokens, known as the anisotropic problem. They analyze the effectiveness of contrastive learning methods and propose to use the Look Neighbors strategy to enhance token representations during training. Experimental results on four WMT datasets demonstrate that our methods consistently improve performance and alleviate the anisotropic problem in conditional masked language models.

**Questions For The Authors:**

1. How many iterations are required to obtain the "CMLM w/ Ours" results in Table 2?


**Reasons To Accept:**

1. The paper contains great motivation by exploring the anisotropic problem in non-autoregressive NMT models.
2. The paper is well organized.
3. Experiments are solid and comprehensive.

**Reasons To Reject:**

1. Lack of comparisons between the proposed method and other models, which also use the CMLM as backbones.
2. Lack of case analysis between different NAT models. It is helpful to understand the effectiveness of the proposed method by comparing outputs generated by different NAT models.


**Reproducibility:**

4: Could mostly reproduce the results, but there may be some variation because of sample variance or minor variations in their interpretation of the protocol or method.

**Reviewer Confidence:**

4: Quite sure. I tried to check the important points carefully. It's unlikely, though conceivable, that I missed something that should affect my ratings.

---

> ### Author Rebuttal · Authors · 2023-08-29
>
> Thanks for your suggestions and comments, we give more explanations for your concerns and questions.
>
> **For your concern 1**
> - In Table 2 of the paper, we compare our method with other representative iterative NAT models which also use CMLM as backbone, such as SMART, Disco, Imputer, CORR and CMLMC. Besides, our methods can also be applied to other CMLM-based models, such as CORR shown in Table 6. This verifys the effectiveness of our methods.
>
> **For your concern 2**
> - Thanks for your advice, next we provide two case analysis which experimented on WMT14 EN-DE dataset to understand the effectiveness of our method better.
>
>   - In Example 1, we can find that the result of CMLM obtains the words "Motorrad rad" which is a grammatical error and the translation of word 'magazine' is missed. In contrast, our methods enhance the dependency of token representations to alleviate above problems.
>   - In Example 2, the words 'ingen' and ',' are repeated several times in the result of CMLM which caused by the anisotropy problem. Our methods are effective to alleviate this problem and provide more fluent translation.
>
>   | Example 1       |                                                              |
>   | --------------- | ------------------------------------------------------------ |
>   | **Source**      | If an affected person has waited for a sufficiently long period of time at a red light , and provided the crossing is clear , they can drive on , explained Stuttgart @-@ based lawyer Ralf Becker in &quot; Motorrad &quot; ( Motorcycle ) magazine. |
>   | **Reference**   | Hat ein Betroffener lange genug an einer roten Ampel gewartet und die Kreuzung ist frei , **kann er weiterfahren , erklärt der Stuttgarter Rechtsanwalt Ralf Becker in der Zeitschrift &quot; Motorrad &quot; .** |
>   | **CMLM**        | Wenn eine betroffene Person eine ausreichend lange Zeit im roten Licht gewartet hat und sofern der Übergang klar ist , kann sie weiterfahren , erklärte der Stuttgarter Rechtsanwalt Ralf Becker im &quot; **Motorrad rad** &quot; ( Motorcycle ) . |
>   | **CMLM w/Ours** | Wenn der Betroffene schon eine ausreichend lange Zeit an einem roten Licht gewartet hat und sofern der übergang klar ist , **kann er weiterfahren , erklärte der Stuttgarter Rechtsanwalt Ralf Becker in der Zeitschrift &quot; Motorrad &quot; .** |
>   |                 |                                                              |
>   | **Example 2**   |                                                              |
>   | **Source**      | Aside from honouring Hugo , the 31st Autumn Festival progressed as usual : Alongside the Betra Male Voice Choir , the Salzstetten Choral Club , the Baisingen Choral Division and the Local Music Society , the Vollmaringen singers delivered a colourful blend of different choral and song styles , which entertained the 400 or so visitors . |
>   | **Reference**   | Neben der Ehrung verlief das 31. Herbstfest in gewohnten Bahnen : Mit dem MGV Betra , dem Liederkranz Salzstetten , der Sängerabteilung Baisingen und dem örtlichen Musikverein hatten die Vollmaringer eine bunte Mischung verschiedener Chöre und Gesangsstile geladen , welche die rund 400 Besucher unterhielten . |
>   | **CMLM**        | Neben der Ehrung von Hugo entwickelte sich das 31. Herbstfest wie üblich : Neben dem Betra Male Voice Chor , dem Chorclub SalzSalzstetten , der Chorabteilung Baisingen ingen und der lokalen Musikgesellschaft gaben die Vollmaringen **ingen ingen ingen** eine farbenfrohe Mischung aus verschiedenen Chor- und Songstilen **, ,** an die etwa 400 Besucher untertierten . |
>   | **CMLM w/Ours** | Neben der Ehren von Hugo schreitet das 31. Herbstfest wie üblich voran : Neben dem Betra Male Voice Chor , dem Salzstetten Chor Club , der Chorabteilung Baisingen und der Local Music Society lieferten die Vollmaringer SänSänger eine farbenfrohe Mischung aus verschiedenen Chor- und Liongstilen , die etwa 400 Besucher begleiteten . |
>
> **For your Question**
> - We obtain all the results with 10 iterations described in Table 2 I_{dec}.

---

### Meta-Review · Area_Chair_3B6L · 2023-09-18

**Recommendation:** 3

**Metareview:**

This paper identifies an important problem in iterative NAR model -- the anisotropic problem where the high similarity of different tokens can mislead the model to generate repetitive tokens at different steps. This finding is inspired by recent work on AR models (Su et al. 2022). The authors revisit the solution from Su et al (2022) -- contrastive learning, and find that it fails to learn the token representation well during training due to the natural difference between NAR and AR models. In response to this problem, the authors propose Look Neighbors to supplement contrastive learning during the training of CMLM. Experiments on widely-used NAR translation benchmarks demonstrate the effectiveness of the proposed approach.

Some necessary information are missing in the submitted version, which are supplemented in the rebuttal. Please include the key information in the revised version:
1. Supplement necessary information about the claims associated with "failure to learn representations" and "anisotropic problem/loss" to make the motivation and design principles more clear.
2. Clarify the motivation of Look Neighbors method, and add an ablation study to prove that the performance improvement is indeed from the method.
3. Experiments on more NAT models are appreciated to demonstrate the universality of the conclusions and the proposed approach.

---

### Decision · Program_Chairs · 2023-10-07

**Decision:**

Accept-Findings

**Comment:**

This paper identifies an important problem in iterative NAR model -- the anisotropic problem where the high similarity of different tokens can mislead the model to generate repetitive tokens at different steps. This finding is inspired by recent work on AR models (Su et al. 2022). The authors revisit the solution from Su et al (2022) -- contrastive learning, and find that it fails to learn the token representation well during training due to the natural difference between NAR and AR models. In response to this problem, the authors propose Look Neighbors to supplement contrastive learning during the training of CMLM. Experiments on widely-used NAR translation benchmarks demonstrate the effectiveness of the proposed approach.

Some necessary information are missing in the submitted version, which are supplemented in the rebuttal. Please include the key information in the revised version:
1. Supplement necessary information about the claims associated with "failure to learn representations" and "anisotropic problem/loss" to make the motivation and design principles more clear.
2. Clarify the motivation of Look Neighbors method, and add an ablation study to prove that the performance improvement is indeed from the method.
3. Experiments on more NAT models are appreciated to demonstrate the universality of the conclusions and the proposed approach.